# Preparation and Characterization of High Amylose Corn Starch–Microcrystalline Cellulose Aerogel with High Absorption

**DOI:** 10.3390/ma12091420

**Published:** 2019-05-01

**Authors:** Qi Luo, Xin Huang, Fei Gao, Dong Li, Min Wu

**Affiliations:** 1College of Engineering, China Agricultural University, P. O. Box 50, No. 17 QinghuaEast Road, Haidian District, Beijing 100083, China; qiluo2128@126.com (Q.L.); fei.gao@chem.ox.ac.uk (F.G.); dongli@cau.edu.cn (D.L.); 2Institute of Environment and Sustainable Development in Agriculture, Chinese Academy of Agricultural Sciences, No. 12 Zhongguancun South Street, Beijing 100081, China; huangxin@caas.cn; 3The Chemistry Research Laboratory, University of Oxford, 12 Mansfield Road OXFORD, Oxford OX1 3TA, UK

**Keywords:** microcrystalline cellulose, high amylose corn starch, dynamic mechanical analyzer

## Abstract

Microcrystalline cellulose (MCC) aerogels were synthesized, blendingwith high amylose corn starch of different contents based on a NaOH–urea solution, and following by vacuum freeze-drying technology. The microstructure of the aerogel was observed by scanning electron microscopy (SEM) as an interconnected, porous three-dimensional structure, while X-ray diffractogram (XRD) measurements showed that the crystalline form was converted from cellulose I to cellulose II during dissolution and regeneration. Thermogravimetric analysis (TGA) showed that the content of starch had little effect on the thermal stability of the aerogel, whereas the content of starch had great influences on absorption and viscoelastic properties. When the ratio of starch was 10% and 15%, the prepared aerogels presented a low density and abundant pores, which endowed the aerogels, not only with the highest absorption ratio of pump oil and linseed oil (10.63 and 11.44 g/g, respectively), but also with better dynamic viscoelastic properties.

## 1. Introduction

With the rapid growth of the global population and rapid economic development, the water pollution caused by the disordered discharge of industrial and domestic wastewater, as well as the leakage caused by the improper treatment of oil, not only cause economic losses, but are also dangerous to the environment. The removal of pollution from water is; therefore, highly desirable. To date, the most commonly used methods to deal with water pollution are physical, chemical, and biological treatments, among which absorption is the simplest and most commonly used physical treatment for its efficiency and cost effectiveness. Among natural materials, the most common absorbents include wheat straw [1], walnut shell [2], and Populus fiber [3], etc. However, they generally have the disadvantage of a poor absorbency and reusability. On the other hand, inorganic aerogels, such as SiO_2_ aerogel [4], graphene oxide aerogel [5] and others, exhibit a high absorption capacity, high porosity, and large specific surface area. However, the degradation of most of inorganic aerogels remains a major environmental challenge and may cause secondary pollution, which greatly limits their application. As a result, novel, green, renewable materials with high absorption that can be degraded by the environment are urgently needed, and cellulose-based aerogels represent an effective alternative.

These materials constitute a new generation of aerogels, which have not only inherited the high porosity and large surface area of preceding aerogels [6], but also have overcome the problem of non-biodegradability of inorganic aerogels [7]. However, cellulose is difficult to dissolve in common solvents, and appropriate solvents are needed to allow dissolution and crosslinking, such as NaOH/urea solution [8,9], N-Methylmorpholin-N-oxide (NMMO) [10,11], or other ionic liquids [12,13]. Supercritical drying [13,14] or vacuum freeze-drying [8,15] are used to avoid the collapse of the aerogels and obtain a solid three-dimensional network structure. In recent years, the research on cellulose-based aerogels has been progressing more and more. Many researchers have aimed to enhance the material’s functions and provide better properties by blending cellulose with other materials such as N, N’-methylene bisacrylamide (MBA)/graphene oxide (GO) hybrid [9], N-methylol dimethylphosphonopropionamide (MDPA) and 1,2,3,4-butanetetracarboxylic acid (BTCA) [16], silica [17], and chitosan [18], etc., but most methods are quite complex. Some studies show that the blend of cellulose and starch can improve the mechanical properties [19], moisture resistance [20], and thermal stability [21] of the composites, whereas there are few reports on the preparation of cellulose aerogel by mixing starch. Compared with raw cellulose, microcrystalline cellulose has the advantage of a high specific surface area, which makes it a material with greater development potential.

Starch is a polysaccharide that can be extracted directly from plants. High amylose starch always has a high resistance to gelatinization and hydrolysis [22,23]. In addition, high amylose starch can form entanglements very easily [24]. The development of polymer blends and composites has been recognized as one of the most effective ways to improve material properties, reduce costs, and expand the range of applications. Therefore, the preparation of aerogels by using cellulose and high amylose starch blending has an important application prospect.

In this paper, we present a new composite aerogel based on microcrystalline cellulose and high amylose corn starch dissolved in a NaOH–urea solution, produced by using vacuum freeze-drying technology. The objective of the study is to investigate the influence on structural characteristics, thermodynamic, absorption, and viscoelastic properties, of charging cellulose-based aerogels with different starch content.

## 2. Materials and Methods

### 2.1. Raw Materials

Microcrystalline cellulose (MCC, about 25 µm) was provided by Sigma-Aldrich (Shanghai) Trading Co., Ltd. (St. Louis, MO, USA). High amylose corn starch (starch, about 12.5 µm) was purchased by Yujing Food Co., Ltd. (Zhengzhou, China) Sodium hydroxide (NaOH) and urea were received from Beijing Chemical Works (Beijing, China). Hydrochloric acid (HCl) was supplied by Sinopharm chemical reagent co. Ltd (Shanghai, China).

### 2.2. Preparation of High Amylose Corn Starch–Microcrystalline Cellulose Hydrogel and Aerogel

Firstly, a ball mill (QM-ISP04, Nanjing University Instrument Plant, Nanjing, China) was used to produced microcrystalline cellulose and high amylose corn starch in smaller particle sizes, about 20 and 11 µm, respectively. This was done by mixing microcrystalline cellulose or high amylose corn starch and ZrO_2_ balls (6 mm in diameter) in a volume ratio of 1:10 or 1:6 for 1 h, which is more conducive to chemical reaction and promotes structural recombination of new materials.

High amylose corn starch–microcrystalline cellulose aerogel samples were prepared utilizing the following methods (Figure 1). Total microcrystalline cellulose and high amylose corn starch solid of 8 g with deionized water of 162 mL were mixed by stirring at 8000 r/min for 10 min (OS20-Pro, Beijing, China), and subjected 3 times to high pressure homogenization (ATS AH 100D, Suzhou, China) at 60 MPa to obtain homogeneous emulsion, then samples were frozen for 2 h. These samples, accordingly, were named as S_0_CA, S_5_CA, S_10_CA, S_15_CA, S_20_CA, S_25_CA, which corresponded to additions of 0%, 5%, 10%, 15%, 20%, and 25% of starch, respectively. Then 14 g NaOH and 24 g urea were added to make the emulsion transparent. After freezing the samples for 24 h, the transparent solid was defrosted and stirred vigorously at atmospheric conditions for 30 min. The white hydrogel was obtained by washing with 5% hydrochloric acid (HCl) and distilled water several times until the leaching solution was neutral. Then the prepared hydrogel was transferred into a freeze-drying machine (LGJ-18C, Beijing, China) to freeze-dry at a temperature of −60 °C and a pressure of 15 Pa for 48 h to fabricate aerogels.

### 2.3. Volume Shrinkage

The volumetric shrinkage of aerogels was measured by the volume change of samples before and after vacuum freeze-drying. The diameter and height of hydrogels before drying were measured, and the volume was calculated, denoted as *V_wet_*. The diameter and height of aerogels after drying were measured, and the volume was calculated, denoted as *V*. Each sample was measured three times and the average value was taken. Volumetric shrinkage *M* (%) was expressed in the following Equation (1):
(1)$$M\left(\%\right)=\frac{V_{wet}-V}{V_{wet}}\times 100$$

### 2.4. Density and Porosity

The mass m_0_ of aerogels was accurately measured by electronic balance (AB204-S), and the volume after drying was calculated. Each sample was measured three times and the average value was taken. The skeletal density of aerogel was calculated by the following Equation (2):
(2)$$\rho =\frac{m_{0}}{V}$$

Ignoring the internal air density of aerogels, ρ_0_ is the bulk density. The porosity *P* (%) was expressed in the following Equation (3):
(3)$$P\left(\%\right)=\left(1-\frac{\rho_{0}}{\rho }\right)\times 100$$

### 2.5. Scanning Electron Microscopy (SEM)

The morphologies of the aerogels were characterized using a scanning electron microscope (SEM) (S-3400N, Hitachi, Tokyo, Japan), operated at an accelerating voltage of 15 kV after being coated with Au.

### 2.6. X-Ray Diffraction (XRD)

Aerogel crystal structures were identified by X-ray diffraction techniques (XRD) (XD-2, Beijing Purkinje General Instrument Co., Ltd., Beijing, China). Scattered radiation was detected in the range of 2θ, from 4° to 40°, with a scanning rate of 2°/min. Crystallinity index I_CI_ (%) was shown in the following Equation (4):
(4)$$I_{CI}\left(\%\right)=\frac{I_{002}-I_{amorph}}{I_{002}}\times 100$$where I_002_ is the maximum intensity of the (002) lattice diffraction and I_amorph_ is the intensity diffraction of the amorphous band [25].

### 2.7. Thermogravimetric Analysis

Aerogel samples were subjected to thermogravimetric analysis (TGA) in a thermogravimetric (TG) analyzer (Q600, TA Instruments, New Castle, DE, USA). All samples were analyzed under a nitrogen atmosphere with flow rate of 100 mL/ min, heating rate of 20 °C/ min, and temperature range of 50–700 °C.

### 2.8. Adsorption Behavior Measurements

Aerogel samples of about 20 mg were immersed into plenty of linseed oil or pump oil. After placing it in the beaker for 24 h to ensure the absorption equilibrium, then the aerogel was put on the sieve for 30 s to remove surface oil, and then was weighed. Each sample was measured three times with an average value, and the adsorption ratio C (%) was measured in the following Equation (5):
(5)$$C\left(\%\right)=\frac{W-W_{0}}{W_{0}}\times 100$$where *W*_0_ and *W* are weights of aerogel sample before and after absorption, respectively.

### 2.9. Creep-Recovery Measurements

Aerogel samples was placed on the dynamic mechanical analyzer (DMA, Q800, TA Instruments, New Castle, DE, USA) to determine the creep curves. The aerogels were cut into 1 cubic centimeter cubes using compression. The preload (0.01N) and the force track (125%) was set and after equilibrating at 30 °C, a stress of 0.0001 MPa was supplied to the aerogels and then the applied stress was removed. The creep time and recovery time were both 10 min. The data of strain was recorded as a function of time.

Burger’s model was used to explain the creep curves. Creep equation of Burger’s model was in the following Equation (6):
(6)$$\varepsilon \left(t\right)=\frac{\sigma_{0}}{E_{M}}+\frac{\sigma_{0}}{E_{K}}\left(1-e^{-t/\tau }\right)+\frac{\sigma_{0}}{\eta_{M}}\cdot t$$where *ε*(*t*) represents the creep strain of the aerogel sample; ***σ*_0_** represents the loaded stress, 0.0001 MPa; *E_M_* and *η_M_* represent the modulus and viscosity of the Maxwell spring and dashpot, respectively; *E_K_* and *η_K_* represent the modulus and viscosity of the Kelvin spring and dashpot, respectively, and *τ* = *η_K_*/*E_K_*; t denotes the time (s) after loading.

### 2.10. Frequency Sweep Test

The mechanical characteristic of the aerogels was measured using a dynamic mechanical analyzer (DMA, Q800, TA Instruments, New Castle, DE, USA). Aerogel samples were pretreated as the creep-recovery measurements. The temperature and the force track were set at 30 °C and 125%. The frequency ranged from 0.1 to 50 Hz and the controlled strain was set at 0.15%. The storage modulus G′, loss modulus G″, and tan δ of the samples were recorded to create the corresponding frequency sweep curves.

### 2.11. Temperature Sweep Test

The mechanical characteristic of the aerogels was measured using a dynamic mechanical analyzer (DMA, Q800, TA Instruments, New Castle, DE, USA). Aerogel samples were pretreated as the creep-recovery measurements. A constant strain of 0.15% (within liner elastic limits) was applied to the samples and the storage modulus G′, loss modulus G″, and tan δ as a function of temperature. The test temperature was increased from 30 to 100 °C at a heating rate of 3 °C/ min with frequency of 1 Hz. The temperature sweep tests were all replicated three times.

## 3. Results and Discussion

### 3.1. Density, Porosity, Volume Shrinkage, and Absorption Capabilities of the Aerogels

A summary of the density, porosity volume shrinkage, and absorption rate of the aerogel samples with different starch content are given in Table 1.

Their density was in a range of 0.125 to 0.151 g cm^−3^, and porosity ranged from 90.1% to 91.8%, which were much better than what was reported in the literature, whereby the cellulose derivatives -based aerogels had high densities (0.25–0.85 g/cm^3^) and low porosities (41%–85%) [26]. The density of commercial porous materials, like Styrofoam [27], and new material aerogels, like Whey protein aerogel [28], were all about 0.1g cm^−3^, which equaled that of the aerogels produced in the present study. As the table showed, there were no significant differences between S_0_CA and S_5_CA of the density and porosity (P > 0.05). However, as a greater amount of starch was added, the density reduced greatly and porosity increased significantly (P < 0.05), compared with S_0_CA, perhaps because of the decrease of the content of microcrystalline cellulose. At the same time, high amylose corn starch can form porous three-dimensional structure to avoid the collapse of microcrystalline cellulose aerogel skeleton.The principle of vacuum freeze-drying is to utilize sublimation to makes the material solvent change from solid to gas directly. Through vacuum freeze-drying, the volume shrinkage rate of S_0_CA was the largest, and compared to the volume shrinkage rate of aerogels with starch were significantly reduced (p < 0.05), indicating that starch can effectively prevent the collapse of aerogels’ skeleton.

When the starch content reached 25%, the density and volume shrinkage rate of aerogels increased, and the porosity decreased, which may be related to the excessive amount of starch, as well as the agglomeration aggravating the collapse of the skeleton.

Oil viscosity is an important factor affecting oil absorbing ability. It has been found that the viscosity of oil could have two different effects on adsorbent adsorption: First, the increase of oil viscosity is more conducive to oil adsorption on the absorbent surface, which increases the absorption ratio; secondly, the increase of oil viscosity will prevent the oil molecules from entering the absorbent, so as to reduce the absorption ratio [29]. The viscosity of linseed oil is 7.4 MPa/s, while the viscosity of vacuum pump oil is 8.0–8.8 MPa/s, and the viscosity of vacuum pump oil is greater than that of linseed oil. It could be seen from the Table 1 that the absorption ratio range of the aerogels obtained to vacuum pump oil was 7.97–10.63 g/g, and that of linseed oil was 8.98–11.44 g/g, which were at the same level as the chitosan/cellulose aerogel [18].

Compared with non-added starch aerogel, the adsorption ratio of the aerogels with different starch all increased significantly (p < 0.05). When the starch content was 15%, the absorption ratio of aerogels to pump oil and linseed oil both reached to the maximum, while the density of aerogels was the minimum and the porosity reached to the maximum according to the Table 1.

### 3.2. Morphologies of the Aerogels

The microstructures of the aerogels with different starch content were examined by SEM (Figure 2). The SEM images of all the aerogels, taken at cross-section, exhibited a three-dimensional porous structure with overlapping layers consisting of interconnected uniform cellulose fibers, which allowed rapid water uptake and showed a high absorption capacity [9,18,30]. The surface of the samples had a staggered texture, which may be caused by the solvent infiltrating into the gel skeleton under the action of gravity and sublimating directly under the condition of freezing, so that it was separated into texture by solvent. In the sample S_0_CA (Figure 2a) and S_5_CA (Figure 2b), due to the excessive content of microcrystalline cellulose, some of cellulose fibers formed a few agglomerations and were thicker. The samples S_10_CA (Figure 2c) and S_15_CA (Figure 2d) showed that with the increase of starch content, the larger the aerogel pore, and the looser the cellulose structure. When the addition of starch reached 20% or more, the cellulose fiber was thickened with starch content and some severe agglomeration of starch attached to the surface of the skeleton, while in S_25_CA (Figure 2f), fiber structures were covered by starch clearly. The phenomenon could be explained by the idea that an appropriate amount of starch could help cellulose fiber to form a three-dimensional porous structure, but an excessive amount of starch could not. Therefore, the starch content should not exceed 15%.

### 3.3. Crystal Structure of the Aerogels

The X-ray diffractogram (XRD) patterns of obtained aerogels with different starch additions, as well as of starch and of microcrystalline cellulose ball-milled for 1 h are shown in Figure 3. The X-ray diffractogram of microcrystalline cellulose ball-milled for 1 h is characteristic of the cellulose I crystal structure (Figure 3) according to literature data [31]. In case of aerogels, the corresponding X-ray diffraction patterns are characteristic to cellulose II [32], due to the dissolving/regeneration process of microcrystalline cellulose in a NaOH–urea aqueous system. As we can see for all aerogels, the characteristic diffraction peaks are observed at Bragg angles of 12°, 20°, and 22°, and are assigned to (110), (110), and (200) crystallographic planes, respectively.

For the XRD patterns of the aerogels, the height of the peak observed at 12°, 20°, and 22° all decreased at first, but once the starch content was over 15%, the height began to increase. According to Table 2, the crystallinity data (I_CI_, refined and calculated by JADE software (JADE 5, Christchurch, New Zealand) showed that with the increasing starch content, the crystallinity of the aerogels first decreased and then increased. When the addition of starch was 15%, the crystallinity reached a minimum of 32.57%. The crystallinity of the sample decreases due to the addition, within aerogels, of starch, with a high content of amorphous region [33]. The cellulose-based aerogel added with starch has a low ordered structure, which is conducive to the diffusion of oil in the aerogel, thus improving the oil adsorption ratio. What is more, there are no significant peaks of starch, and it indicated starch in the aerogels was amorphous.

### 3.4. Thermal Properties of the Aerogels

Thermogravimetric (TG) curves and derivative thermogravimetric (DTG) curves of microcrystalline cellulose, high amylose corn starch, and the aerogels with different additions of starch are presented in Figure 4. The starch and MCC showed a small weight loss at the beginning of heating that could correspond to the evaporation of moisture [34]. However, there was no obvious weight loss in aerogels obtained, which inferred that starch and MCC were both hydrophilic, but the aerogels formed after MCC and starch dissolution and regeneration lowered the hygroscopicity. It can be seen that there was a weight loss from about 100% to 70% in the temperature of 150 to 230 °C due to the removal of unreactive urea left in the all aerogels [35]. From the DTG curves, it was clearly found that a weight loss occurred in the range of 280–320 °C due to the degradation and burning of the starch, and in the range of 320–360 °C due to the degradation and burning of the MCC. When the samples were subjected to heating from 50 to 700 °C under a nitrogen atmosphere, the residual weight of MCC and starch were 2.85% and 10.73%, respectively, which were much lower than other aerogels. About aerogels, it can be seen from the figure that the residues of S_0_CA was the highest, about 39.73%, while S_15_CA was the lowest, about 34.15%. This may be because when the content of starch was 15%, the network of aerogels through hydrogen bonds was more orderly and the heat transfer was more uniform, attributing more thorough combustion. The onset temperature of all aerogels was much lower than MCC and starch, indicating that the dissolution and regeneration of MCC and starch contributed to reducing the thermal stability. However, there were no obvious differences between pure cellulose-based aerogel and the aerogels with different starch content, revealing that the introduction of starch did not enhance the cellulose skeleton stability to some degree.

### 3.5. Dynamic Mechanical Properties of the Aerogels and Burger’s Model Analysis

The creep and creep-recovery strains as a function of time of the aerogels prepared with different starch content were presented in Figure 5. The strain curves of the sample aerogels showed typical creep characteristics and showed a similar trend over time. The creep deformation curves can be seen as three stages: The first stage shows that the strain of aerogels increase rapidly in a short time; the second stage shows the delayed deformation and the strain of aerogels increased slowly with the time; the last stage shows strain recovery after the external force is removed [36]. The strain rose sharply at first and then rose gently. In the recovery phase it is just the opposite. When starch content did not exceed 20%, the creep strain curves were significantly lower than that of S_0_CA, indicating that the starch was distributed well in the cellulose-base aerogels and the addition of starch enhanced the creep resistance. S_10_CA presented the lowest creep strain, demonstrating that the aerogels reached the best creep resistance. However, when the addition of starch exceeded 20%, the creep strain curves were significantly higher than the other aerogels. This was probably due to excess starch that could not be distributed in the cellulose-based aerogels uniformly; it was confirmed that some severe agglomeration of starch attached to the surface from Figure 2e,f.

The creep behavior can be described by a four-element Burgers model [37]. The four parameter values (E_M_, E_K_, τ, and η_M_) and the determination coefficient R^2^ obtained from the Burgers model were fitted as shown in Table 3. It can be seen that all the parameters had the coefficient of determination (R^2^) ≥ 0.989, suggesting that the four-element Burgers model can represent the creep behavior of the aerogels well.

E_M_ represents an instantaneous creep deformation that can be restored immediately when the external force is withdrawn. The greater the value, the better the elasticity of the material [38]. The tabulated values also show that the value of E_M_ increased when the starch content increased from 0% to 15%, whereas it decreased within 20% to 25%, indicating that excessive starch could destroy the viscosity of cellulose-based aerogels. When the content of starch reached 15%, the elasticity of aerogels was the best. E_K_ represents the retarded elastic modulus and τ is the retardation time of the Kelvin component. The greater the values, the stickier the material. For S_10_CA, the value of E_K_ reached to maximum; however, τ reached to minimum. When the starch content reached 15%, the value of E_K_ and τ were both high relatively. η_M_ represents the viscous component of the dashpot. S_15_CA showed the maximum η_M_ value of 1.777 ± 0.359MPa·s^−1^, which meant that S_15_CA had the least irreversible variables and better viscoelasticity. However, once the starch content reached 20% or more, the value of η_M_ decreased significantly and even was lower than S_0_CA.

Through the above chart analysis, it shows that the incorporation of high amylose corn starch could improve the viscoelasticity and creep resistance of cellulose-based aerogels. Similarly, the cellulose-based aerogels blended with the starch content of 15% presented the best viscoelasticity.

### 3.6. Frequency Sweep Properties of the Aerogels

The variation of the storage modulus (G′), the loss modulus (G″), and tan δ as a function of frequency with the different starch content range is represented in Figure 6. As can be seen from the figure, the general trend of G′, G″, and tan δ increased as the frequency increased. The values of G′ were much greater than the values of G″ throughout the frequency range, which indicates that the elastic property of this material is better than that of viscous property. When adding 10% content starch, the maximum values of G′, G″, and tan δ of the aerogels obtained were all reached, which demonstrated that adding the starch at a suitable content can improve the mechanical properties of the cellulose-based aerogels. The lowest G′ and G″ were observed at the starch content value of 20% and 25%, respectively, within the entire frequency range. Adding too much starch could lead to electrostatic agglomeration and excess starch was not evenly distributed in the aerogel.

### 3.7. Temperature Sweep Properties of the Aerogels

Figure 7 shows the variation of storage modulus G′ and loss modulus G″ as a function of temperature within the tested starch content range. As can be seen from the graph, with the increase of temperature G′, values increased in the beginning and then decreased. It may be because at lower temperature, the high porosity of aerogels caused the aerogels to become dehydrated and the storage modulus increased. When heated, aerogels began to adsorb moisture in air and became soft and easy to form, and the storage modulus decreased. When the starch content was 20% and 25%, the storage modulus was the smallest, indicating that the aerogel stiffness was the lowest at this time. When the starch content were 10% and 15%, the storage modulus was larger than that of the aerogel without starch, which indicated that adding suitable starch to the material had better stiffness characteristics and effectively inhibited the movement of the polymer chain caused by the ideal dispersant.

The temperature sweep curves of the aerogel samples showed similar profiles in different starch content, all of which first increased and then decreased, and reached the maximum value between 50–70 °C. The softening of materials can be explained by molecular motion in free volume theory [38]. The molecular motion of the polymer chain is fixed, and the polymer is in the glassy state at lower temperature, which made the samples hard and rigid. When the temperature was raised, enough heat was supplied to the polymer, which increased the molecular activity and they could slide against one another. The polymer will gradually change to a flexible and deformable rubber state [39]. Generally, the temperature corresponding to the loss modulus peak in the DMA curve is Tg, and the glass transition temperature of aerogel samples with starch content of 0%, 5%, 10%, 15%, 20%, and 25% is not obvious. It shows that the glass transition temperature of aerogels does not change with the amount of starch added.

## 4. Conclusions

In this study, a low density, high porosity, and low-cost aerogel with high absorption capability has been successfully synthesized using MCC and starch, through a vacuum freeze-drying process. SEM analysis revealed an interconnected, highly porous three-dimensional structure of the obtained aerogel, while XRD measurements showed a transformation of the crystal form from cellulose I to cellulose II during the dissolution and regeneration process. The weight ratio of starch had a great influence on the aerogel’s properties. When the ratio was 10% and 15%, the prepared aerogel presented a low density and abundant pores, which endowed the material not only with the highest absorption ratio of pump oil and linseed oil (10.63 and 11.44 g/g, respectively), but also with excellent dynamic viscoelastic properties. In conclusion, the study has identified the suitable starch content range for fabricating operable, efficient, and cost-efficient aerogels.

## Figures and Tables

**Figure 1 materials-12-01420-f001:**
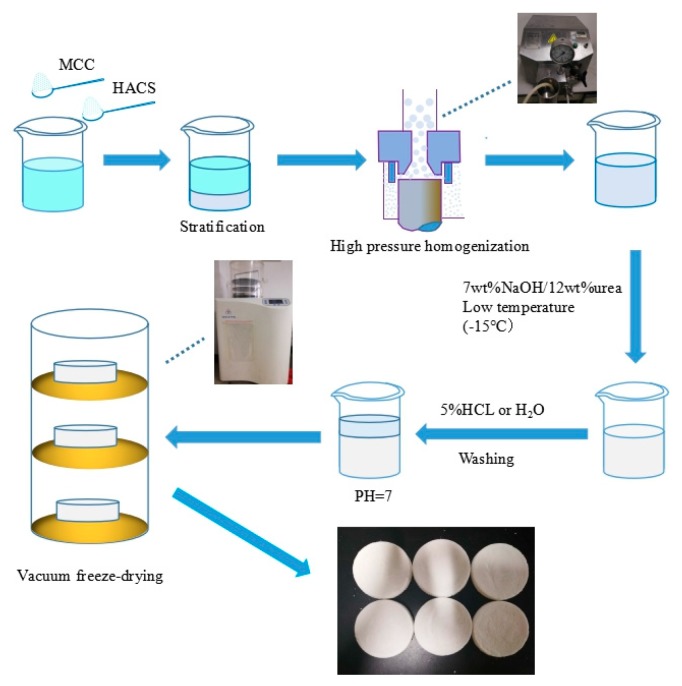
Hydrogel and aerogel preparation process.

**Figure 2 materials-12-01420-f002:**
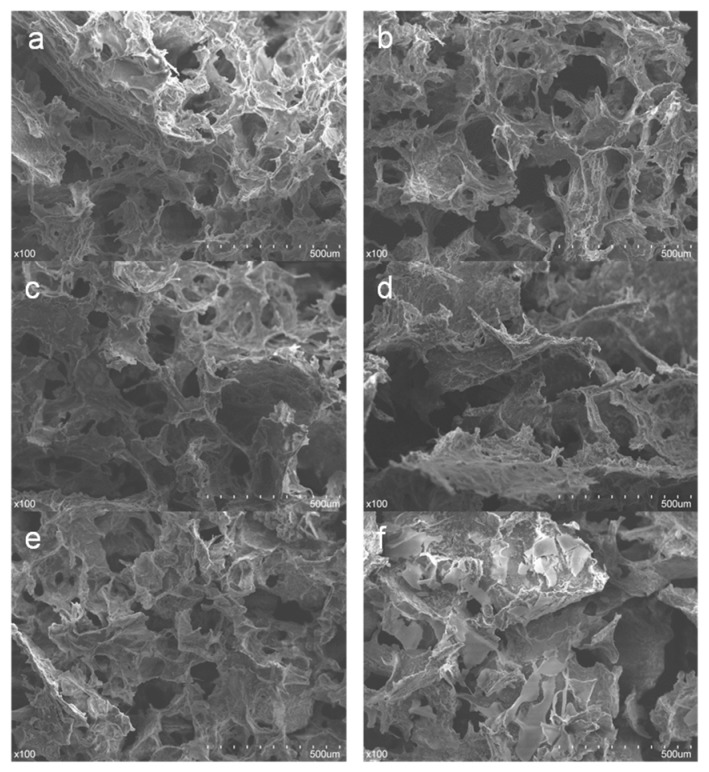
SEM images of the aerogels with different starch content (**a**: S_0_CA, **b**: S_5_CA, **c**: S_10_CA, **d**: S_15_CA, **e**: S_20_CA, **f**: S_25_CA).

**Figure 3 materials-12-01420-f003:**
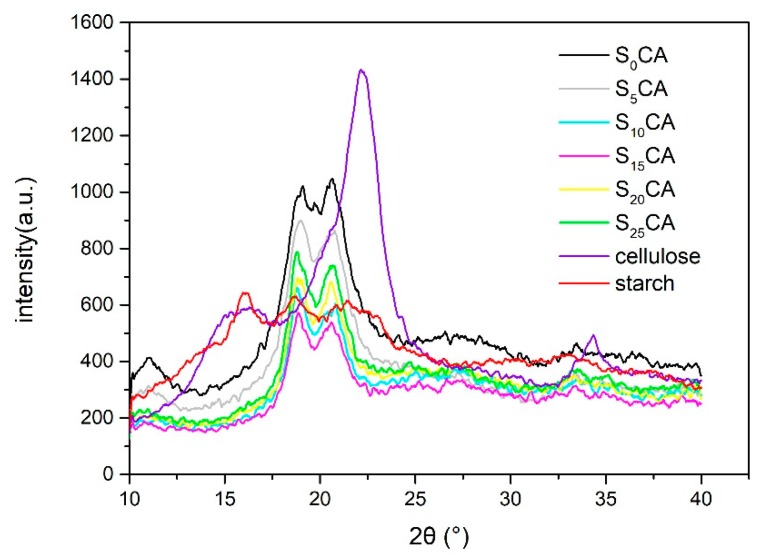
X-ray diffractogram (XRD) pattern of the aerogels with different starch content.

**Figure 4 materials-12-01420-f004:**
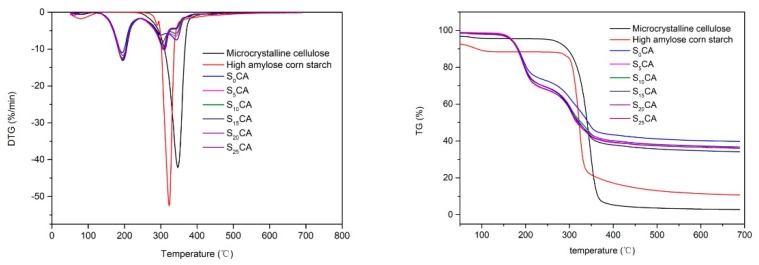
The TG and DTG curves of MCC, starch, and the aerogels with different starch content.

**Figure 5 materials-12-01420-f005:**
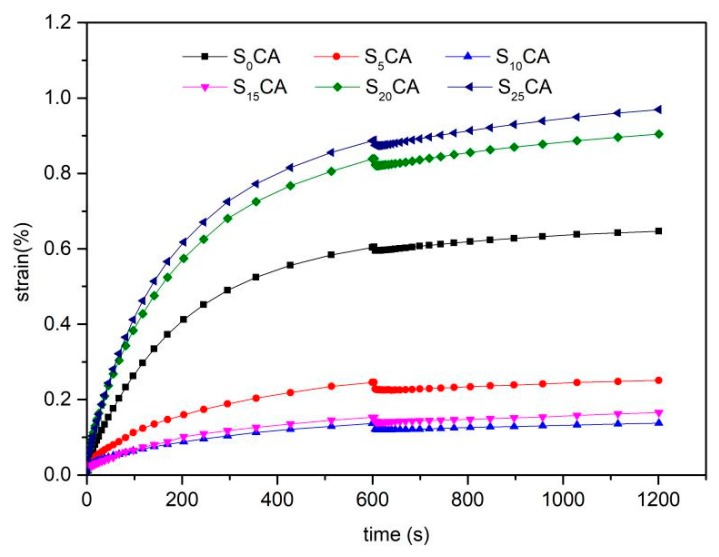
Creep versus time curves of the aerogels with different starch content.

**Figure 6 materials-12-01420-f006:**
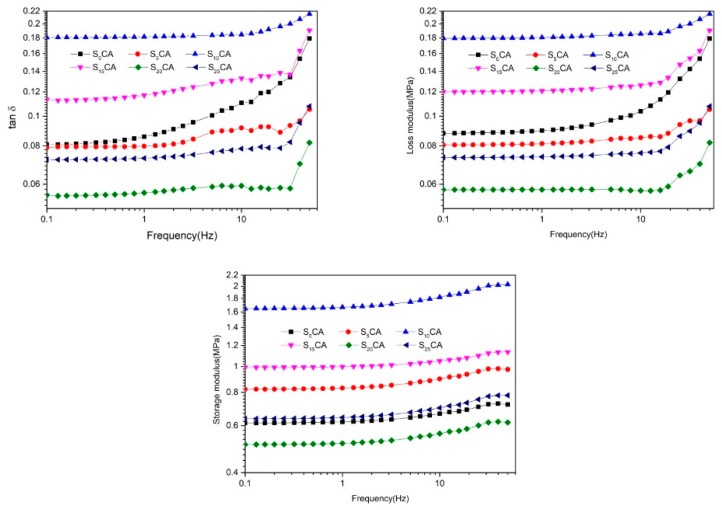
Storage modulus, loss modulus, and tan δ versus frequency curves of the aerogels with different starch content.

**Figure 7 materials-12-01420-f007:**
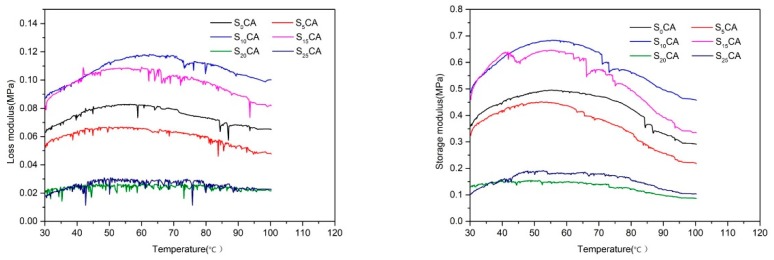
Storage modulus and loss modulus versus temperature curves of the aerogels with different starch content.

**Table 1 materials-12-01420-t001:** Density (ρ), porosity (P), volume shrinkage (S), and absorption rate of the aerogels with different starch content.

Sample	ρ/gcm^−3^	P (%)	S (%)	Absorption Ratio (g/g)	
				Vacuum Pump Oil	Linseed Oil
**S_0_CA**	0.149 ± 0.001 ^c^	90.2 ± 0.1 ^a^	44.5 ± 0.8 ^b^	7.97 ± 0.42 ^a^	8.98 ± 0.20 ^a^
**S_5_CA**	0.151 ± 0.003 ^c^	90.1 ± 0.2 ^a^	35.9 ± 2.6 ^a^	9.05 ± 0.27 ^b^	9.13 ± 0.24 ^a^
**S_10_CA**	0.132 ± 0.005 ^ab^	91.4 ± 0.3 ^bc^	38.6 ± 2.5 ^a^	9.24 ± 0.42 ^b^	10.00 ± 0.64 ^bc^
**S_15_CA**	0.125 ± 0.006 ^a^	91.8 ± 0.4 ^c^	37.9 ± 0.9 ^a^	10.63 ± 0.43 ^d^	11.44 ± 0.36 ^d^
**S_20_CA**	0.128 ± 0.002 ^a^	91.6 ± 0.1 ^c^	37.7 ± 0.7 ^a^	9.94 ± 0.45 ^c^	10.55 ± 0.54 ^c^
**S_25_CA**	0.135 ± 0.003 ^b^	91.1 ± 0.2 ^b^	42.1 ± 0.8 ^b^	8.86 ± 0.28 ^b^	9.47 ± 0.46 ^ab^

* These samples were named S_0_CA, S_5_CA, S_10_CA, S_15_CA, S_20_CA and S_25_CA, respectively corresponding to 0%, 5%, 10%, 15%, 20% and 25% of the content of starch. * Values represent mean ± the standard deviation. a–d in the same column with different superscripts are significantly different (p < 0.05).

**Table 2 materials-12-01420-t002:** Crystallinity (I_CI_) value of aerogel with different starch content.

Samples	S_0_CA	S_5_CA	S_10_CA	S_15_CA	S_20_CA	S_25_CA
I_CI_/%	49.61 ± 1.59 ^e^	42.37 ± 1.42 ^d^	34.72 ± 0.54 ^b^	32.57 ± 1.33 ^a^	36.31 ± 0.30 ^b^	33.09 ± 0.40 ^c^

* Values represent mean ± the standard deviation. a–e in the same column with different superscripts are significantly different (p < 0.05).

**Table 3 materials-12-01420-t003:** The parameters of Burger’s models for the aerogels with different starch content.

Sample	*E_M_*(MPa)	*E_K_*(MPa)	*τ*(s)	*η_M_*(MPa·s^−1^)	R^2^
**S_0_CA**	0.475 ± 0.014 ^d^	0.021 ± 0.001 ^a^	2.562 ± 0.405 ^d^	0.888 ± 0.151 ^b^	0.999
**S_5_CA**	0.415 ± 0.036 ^c^	0.082 ± 0.006 ^b^	1.831 ± 0.221 ^b^	0.974 ± 0.091 ^b^	0.995
**S_10_CA**	0.593 ± 0.037 ^e^	0.208 ± 0.007 ^d^	1.058 ± 0.042 ^a^	1.344 ± 0.135 ^c^	0.989
**S_15_CA**	0.662 ± 0.027 ^f^	0.120 ± 0.029 ^c^	2.215 ± 0.211 ^c^	1.777 ± 0.359 ^d^	0.995
**S_20_CA**	0.236 ± 0.024 ^a^	0.017 ± 0.003 ^a^	2.101 ± 0.101 ^bc^	0.430 ± 0.079 ^a^	0.999
**S_25_CA**	0.340 ± 0.034 ^b^	0.016 ± 0.003 ^a^	2.081 ± 0.060 ^bc^	0.450 ± 0.038 ^a^	0.999

* *E_M_* and *η_M_* represent the modulus and viscosity of the Maxwell spring and dashpot, respectively; *E_K_* and *η_K_* represent the modulus and viscosity of the Kelvin spring and dashpot, respectively, and *τ = η_K_/E_K_*; t denotes the time (s) after loading. R^2^ represents coefficient of determination. * Values represent mean ± the standard deviation. a–f in the same column with different superscripts are significantly different (p < 0.05).

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
