# Peer review of "Preparation and Characterization of High Amylose Corn Starch–Microcrystalline Cellulose Aerogel with High Absorption"

_materials, 2019, doi:10.3390/ma12091420_

Round 1
Reviewer 1 Report
In the paper "Preparation and characterization of high amylose corn 3 starch/microcrystalline cellulose aerogel with high 4 absorption" authors investigate the best ratio of microcristalline cellulose/high amylose starch to manufacture aerogels for absorbing oils.
Overall the work is well presented. Introduction offers a good background on the topic and methods have enough information for other researchers to replicate the experiments.
Authors offer a good amount of results and conclusions drawn thereof are well supported.
Therefore this reviewer recommends the publication of this work in its current form.
Author Response
Detailed Response to Editors and Reviewers
Paper submitted to Materials Journal
Manuscript ID: materials-480708
Title: Preparation and characterization of high amylose corn starch/microcrystalline cellulose aerogel with high absorption
Dear Editors and Reviewers:
Thank you for your letter and for the reviewer’s comments on our manuscript entitled “Preparation and characterization of high amylose corn starch/microcrystalline cellulose aerogel with high absorption” (Materials). Those comments are all valuable and very useful to perfect our manuscript and improve our paper, as well as the important guiding significance to our researches. We have studied comments carefully and have revised the manuscript according to your suggestion which we hope to meet with approval.
We would like to express our great appreciation for your comments on our paper. We responded point by point to each reviewer comments as listed below, along with a clear indication of the location of the revision.
To Reviewer 1:
Comment 1: In the paper "Preparation and characterization of high amylose corn starch/microcrystalline cellulose aerogel with high absorption" authors investigate the best ratio of microcrystalline cellulose/high amylose starch to manufacture aerogels for absorbing oils.
Overall the work is well presented. Introduction offers a good background on the topic and methods have enough information for other researchers to replicate the experiments.
Authors offer a good amount of results and conclusions drawn thereof are well supported.
Therefore this reviewer recommends the publication of this work in its current form.
Authors Response: We sincerely thank you for taking time to read the article and give us your scientific comments.
We again sincerely thank you for your time to go through this manuscript, and sincerely hope that our response is adequate. Thank you!
Best regards,
Min Wu, Ph D Associate professor
Assistant Dean of the College of Engineering
PO Box 50 China Agricultural University (East Campus) Qinghua Donglu 17, Haidian District Beijing 100083 P R of China
Tel.: 86 10 62736883, 13810328724
Fax: 86 10 62736883
Email: [email protected]
Reviewer 2 Report
The manuscript materials-480708 describes the preparation and characterization of aerogels based on microcrystalline cellulose-amylose corn starch.
Recently, it was proposed that the synthesis of aerogels imply polymer dissolution, solution gelation, the solvent exchange and drying with supercritical carbon dioxide!
There is an agreement in the scientific area that aerogels are open pores solid networks with high porosity (at least 90%) and high specific surface area!
Different ways are used to prepare three-dimensional polysaccharide-based materials (aerogels, cryogel, xerogels), but most of the drying methods do not lead to a mesoporous matter, i.e. with high specific surface area.
Thus, in my opinion, the authors obtained hydrogels and then after freeze-drying, cryogels!!
Moreover, the main parameters which characterize aerogels texture are the Specific surface area (SBET) and pore size distribution, which are not discussed in this manuscript!!!
Abstract
p. 1, L. 17. “make full use of low-cost raw materials”! How do the authors appreciate low-cost raw materials? In my opinion, microcrystalline cellulose isn’t a low-cost raw material!
p. 1. L. 17-18. The authors referred to “simplify the preparation process”. Simplify in compare with what?
The authors mixed cellulose with amylose to simplify the method?
Introduction
p. 2, L. 57-61. The examples regarding the copolymerization of cellulose with starch is not related to the subject of the manuscript! There are a lot of data related to starch-based aerogels!
2.2. Preparation of high amylose corn starch/microcrystalline cellulose hydrogel and aerogel
p. 2, L. 90-91. “The White hydrogel was obtained by replacing 5% hydrochloric acid (HCl) and distilled water to neutral”? This should be the regeneration/washing step in the obtaining of the hydrogels! The phrase needs to be reorganized!
2.4. Density and porosity
p. 3, L. 108. “purification ?0 is the density of cellulose/starch skeleton”?
2.8. Adsorption behaviour measurements
p. 4, L. 127-128. “the aerogel was salvaged to make it free to drop 30s and”?
3.1. Density, porosity volume shrinkage and absorption capabilities of the aerogels
The explanation regarding the network formation is ambiguous!
p. 5, L. 163-165. Why the authors compare them samples with “polysaccharide derivatives-based aerogels”? There is no reason for this!
p. 5, L. 173-174. Do the authors consider amylose corn starch as cross-linking agent? Did they use amylose in the network as cross-linking agent?
3.2. Morphologies of the aerogels
As I already mentioned, the important parameters which should be presented are the pore sizes and the pore size distribution for each aerogel!
p. 6, L. 210-12. “The sample S10CA (Fig. 2c) and S15CA (Fig. 2d) showed that with the increase of starch addition and the decrease content of microcrystalline cellulose, the larger the holes of aerogels were, the looser the structure of cellulose was”? The phrase needs to be reorganized!
3.3. Crystal structure of the aerogels
p. 7, L. 219. All the diffractograms from Figure 3 should be improved by eliminating the noise and obtaining of smooth lines, in order to observe the differences between the crystalline structure of the samples!
p. 7, L. 224-226. “According to the literature, cellulose conform to the cellulose I crystal structure [28], indicating that microcrystalline cellulose dissolving in NaOH/ urea solvent and regeneration could change the crystal structure of cellulose”? This sentence is ambiguous!
p. 7, L. 224-226. “This may be because starch consumed the hydroxyl group of cellulose, leading to the weakening of hydrogen bonds between cellulose molecules, the rearrangement of cellulose molecules, and the change of cellulose layer spacing”? This sentence is ambiguous!
The starch can’t consume the hydroxyl group of cellulose! There is no prove for the rearrangement of cellulose molecules!
p. 7, L. 234-235. “What’s more, there are no significant peaks of starch, and it indicated starch in the aerogels was amorphous”? There is no XRD for amylose which could underline this assumption!
All this paragraph should be reorganized!
4. Conclusions
“XRD measurements showed a crystal form transformation from cellulose I to cellulose II during the progress of dissolution and regeneration”? This is not the reason why XRD was used in this study! Moreover, there is no prove of the transformation from cellulose I to cellulose II!
In conclusion, in the Introduction part are only general information, not all the time related to the aim of the manuscript, in the Results part it is not clearly established the influence of each component on the structures of aerogels and neither how the 3D network was established! Overall there are a lot of unclear assumptions!
Thus, in my opinion, the data presented in the manuscript need improvements and the paper should be rejected.
Author Response
Detailed Response to Editors and Reviewers
Paper submitted to Materials Journal
Manuscript ID: materials-480708
Title: Preparation and characterization of high amylose corn starch/microcrystalline cellulose aerogel with high absorption
Dear Editors and Reviewers:
Thank you for your letter and for the reviewer’s comments on our manuscript entitled “Preparation and characterization of high amylose corn starch/microcrystalline cellulose aerogel with high absorption” (Materials). Those comments are all valuable and very useful to perfect our manuscript and improve our paper, as well as the important guiding significance to our researches. We have studied comments carefully and have revised the manuscript according to your suggestion which we hope to meet with approval.
We would like to express our great appreciation for your comments on our paper. We responded point by point to each reviewer comments as listed below, along with a clear indication of the location of the revision.
To Reviewer 2:
Recently, it was proposed that the synthesis of aerogels imply polymer dissolution, solution gelation, the solvent exchange and drying with supercritical carbon dioxide!
There is an agreement in the scientific area that aerogels are open pores solid networks with high porosity (at least 90%) and high specific surface area!
Different ways are used to prepare three-dimensional polysaccharide-based materials (aerogels, cryogel, xerogels), but most of the drying methods do not lead to a mesoporous matter, i.e. with high specific surface area.
Thus, in my opinion, the authors obtained hydrogels and then after freeze-drying, cryogels!!
Comment 1: Moreover, the main parameters which characterize aerogels texture are the Specific surface area (SBET) and pore size distribution, which are not discussed in this manuscript!!!
Authors Response: Thanks for your advice, which is very important. The specific surface area (SBET) and pore size distribution are the main parameters to characterize the aerogel structure. The purpose of our study is to make full use of low-cost raw materials, and simplify the preparation process, so as to obtain aerogel-like products prepared by freeze-drying in industrial production. After performance testing, it is proved that these products have certain adsorption and aerogel characteristics. Thanks to your advice, I have found my shortcomings in my current work. we will supplement the research on specific surface area and porosity of aerogel and conduct in-depth research on the preparation process of different drying methods in our future studies.
Abstract
Comment 2: p. 1, L. 17. “make full use of low-cost raw materials”! How do the authors appreciate low-cost raw materials? In my opinion, microcrystalline cellulose isn’t a low-cost raw material!
Authors Response: Thanks for your comments and suggestion. In this paper, Aerogels were prepared by mixing microcrystalline cellulose with starch. We hope to reduce the cost of aerogels and improve their properties by adding cheap starch. What’s more, we will consider the direct extraction of cellulose from agricultural waste to further reduce the cost.
Comment 3: p. 1. L. 17-18. The authors referred to “simplify the preparation process”. Simplify in compare with what?
The authors mixed cellulose with amylose to simplify the method?
Authors Response: Thanks for your suggestions. Many researchers aimed to enhance the functions and improve the properties of cellulose-based aerogels by blending cellulose with other materials, such as silica ( Fu, J.J., Wang, S.Q., He,C.X., Lu, Z.X., Huang, J.D., Chen,Z.L. Facilitated fabrication of high strength silica aerogels using cellulose nanofibrils as scaffold. Carbohydrate Polymers. 2016, 147, 89-96.), N-methylol dimethylphosphonopropionamide (MDPA) and 1,2,3,4-butanetetracarboxylic acid (BTCA) (Guo, L., Chen, Z., Lyu, S., Fu, F., & Wang, S. Highly flexible cross-linked cellulose nanofibril sponge-like aerogels with improved mechanical property and enhanced flame retardancy. Carbohydrate Polymers. 2018, 179, 333-340), but most methods are more complex. The addition of starch can not only simplify the process but also improve the properties of aerogel. Explanations were added at lines 54-56 of the introduction.
Introduction
Comment 4: p. 2, L. 57-61. The examples regarding the copolymerization of cellulose with starch is not related to the subject of the manuscript! There are a lot of data related to starch-based aerogels!
Authors Response: Thanks for your valuable suggestions. This section has been deleted and we have added the reason for the preparation of aerogel by mixing starch with cellulose. The revised details can be found in Line 59-61.
2.2. Preparation of high amylose corn starch/microcrystalline cellulose hydrogel and aerogel
Comment 5: p. 2, L. 90-91. “The White hydrogel was obtained by replacing 5% hydrochloric acid (HCl) and distilled water to neutral”? This should be the regeneration/washing step in the obtaining of the hydrogels! The phrase needs to be reorganized!
Authors Response: Thanks for your suggestions, we have now revised and refined our sentences and paragraphs according to your suggestions. The White hydrogel was obtained by washing with 5% hydrochloric acid (HCl) and distilled water for several times until the leaching solution was neutral. Now we have changed this content to the MS. (Line 88-90 in the new revised manuscript)
2.4. Density and porosity
Comment 6: p. 3, L. 108. “purification ?0 is the density of cellulose/starch skeleton”?
Authors Response: Thanks for your suggestions. ?0 is the density of cellulose and starch in a certain proportion.
2.8. Adsorption behavior measurements
Comment 7: p. 4, L. 127-128. “the aerogel was salvaged to make it free to drop 30s and”?
Authors Response: Thanks for your comments. After salvaging the aerogel from the oil, the oil on the surface would affect the determination of the adsorption properties of the aerogel, so it was allowed to drip freely for 30s to remove the oil on the surface.
3.1. Density, porosity volume shrinkage and absorption capabilities of the aerogels
Comment 8: The explanation regarding the network formation is ambiguous!
Authors Response: Thanks for your suggestions. We have referred to a lot of literature, cellulose dissolved in sodium hydroxide/urea solution will form a three-dimensional network structure such as Geng, H. Preparation and characterization of cellulose/ N,N’ -methylene bisacrylamide/graphene oxide hybrid hydrogels and aerogels. Carbohydrate Polymers. 2018, 196, 289-298., Li, Z., Shao, L., Hu, W., Zheng, T., Lu, L., Cao, Y., & Chen, Y. Excellent reusable chitosan/cellulose aerogel as an oil and organic solvent absorbent. Carbohydrate Polymers. 2018, 191, 183-190, and so on. We have added two papers to confirm this statement in Line 205. At the same time, the network of aerogel can be seen directly by SEM. Density, porosity and volume shrinkage are parts of the adsorption property of aerogel.
Comment 9: p. 5, L. 163-165. Why the authors compare them samples with “polysaccharide derivatives-based aerogels”? There is no reason for this!
Authors Response: Thanks for your suggestions. We are very sorry that we didn’t make it clear due to our incorrect expression. The words “polysaccharides derivatives-based aerogels” have been changed to “cellulose derivatives-based aerogels” in Line 165. Both cellulose and starch are polysaccharides and we compared the samples with cellulose derivatives-based aerogels to illustrate that the aerogels we prepared had lower densities and higher porosities.
Comment 10: p. 5, L. 173-174. Do the authors consider amylose corn starch as cross-linking agent? Did they use amylose in the network as cross-linking agent?
Authors Response: Thanks for your suggestions. We are very sorry for our incorrect writing. In this paper, we prepared a kind of cellulose-based aerogel blending with starch. Starch could help cellulose-based aerogels achieve more stable three-dimensional structures and avoid aerogel skeleton of cellulose collapse. Limited studies have explored the interaction and compatibility between high amylose corn starch and microcrystalline cellulose. Therefore, we don't have enough evidence to consider starch as a crosslinking agent. We have removed the content that starch is a crosslinking agent in Line 173-175.
3.2. Morphologies of the aerogels
Comment 11: As I already mentioned, the important parameters which should be presented are the pore sizes and the pore size distribution for each aerogel!
Authors Response: We sincerely thanks for your suggestions. We will refine this question in future studies.
Comment 12: p. 6, L. 210-12. “The sample S10CA (Fig. 2c) and S15CA (Fig. 2d) showed that with the increase of starch addition and the decrease content of microcrystalline cellulose, the larger the holes of aerogels were, the looser the structure of cellulose was”? The phrase needs to be reorganized!
Authors Response: Thanks for your suggestions, we have now revised and refined our sentences and paragraphs according to your suggestions. The phrase has been changed to “The sample S10CA (Fig. 2c) and S15CA (Fig. 2d) showed that with the increase of starch content and the decrease of microcrystalline cellulose content, the larger the aerogel pore, the looser the cellulose structure”. We have changed this content to the MS. (Line 211-212 in the new revised manuscript)
3.3. Crystal structure of the aerogels
Comment 13: p. 7, L. 219. All the diffractograms from Figure 3 should be improved by eliminating the noise and obtaining of smooth lines, in order to observe the differences between the crystalline structure of the samples!
Authors Response: We sincerely thank you give us your advice. The diffractograms were improved by eliminating the noise and obtaining of smooth lines in revised manuscript in Fig. 3. The detailed revision can be found in Line 219.
Comment 14: p. 7, L. 224-226. “According to the literature, cellulose conform to the cellulose I crystal structure [28], indicating that microcrystalline cellulose dissolving in NaOH/ urea solvent and regeneration could change the crystal structure of cellulose”? This sentence is ambiguous!
Authors Response: Thanks for your suggestions. We have added XRD images of the microcrystalline cellulose with 1h ball milling in Fig. 3. to prove that microcrystalline cellulose dissolving in NaOH/ urea solvent and regeneration could change the crystal structure of cellulose from cellulose Ⅰto cellulose Ⅱ. We have changed this content to the MS. (Line 219 in the new revised manuscript).
Comment 15: p. 7, L. 224-226. “This may be because starch consumed the hydroxyl group of cellulose, leading to the weakening of hydrogen bonds between cellulose molecules, the rearrangement of cellulose molecules, and the change of cellulose layer spacing”? This sentence is ambiguous!
The starch can’t consume the hydroxyl group of cellulose! There is no prove for the rearrangement of cellulose molecules!
Authors Response: Thanks for your suggestions. We are very sorry for our incorrect writing. From the article of Barouni E (2015) (Barouni, E, Petsi, T, Kanellaki, M, Bekatorou, A, & Koutinas, A. Tubular cellulose/starch gel composite as food enzyme storehouse. Food Chemistry. 2015, 188, 106-110.), Crystallinity indices decreased because the surface corresponding to amorphous regions increased. We have changed this content to the MS. (Line 232-235 in the new revised manuscript).
Comment 16: p. 7, L. 234-235. “What’s more, there are no significant peaks of starch, and it indicated starch in the aerogels was amorphous”? There is no XRD for amylose which could underline this assumption!
Authors Response: Thanks for your suggestions. We have added XRD images of the amylose with 1h ball milling in Fig. 3. to prove that there is XRD for amylose. We have changed this content to the MS. (Line 219 in the new revised manuscript).
4. Conclusions
Comment 17: “XRD measurements showed a crystal form transformation from cellulose I to cellulose II during the progress of dissolution and regeneration”? This is not the reason why XRD was used in this study! Moreover, there is no prove of the transformation from cellulose I to cellulose II!
Authors Response: Thanks for your suggestions. In this paper, we studied the XRD of the aerogels to find out whether the crystal structure of the aerogels changed. We have added XRD images of the microcrystalline cellulose with 1h ball milling in Fig. 3. to prove of the transformation from cellulose I to cellulose II. We have changed this content to the MS. (Line 219 in the new revised manuscript).
Comment 18: In conclusion, in the Introduction part are only general information, not all the time related to the aim of the manuscript, in the Results part it is not clearly established the influence of each component on the structures of aerogels and neither how the 3D network was established! Overall there are a lot of unclear assumptions!
Authors Response: Thanks for your suggestions. We sincerely thank you give us your scientific comments. According to your suggestion, we have made some revises to the article.
We again sincerely thank you for your time to go through this manuscript, and sincerely hope that our response is adequate. Thank you!
Best regards,
Min Wu, Ph D Associate professor
Assistant Dean of the College of Engineering
PO Box 50 China Agricultural University (East Campus) Qinghua Donglu 17, Haidian District Beijing 100083 P R of China
Tel.: 86 10 62736883, 13810328724
Fax: 86 10 62736883
Email: [email protected]
Reviewer 3 Report
The authors have described the experiments, data and deductions in a clear and concise matter. What helped the authors were that the experimental data did not show any deviation or anomaly. Hence, the explanations were simple to understand and easy to deduce. Oveall, I believe that this work is sound and worth publishing. Some minor editing and language correction is however required.
Author Response
Detailed Response to Editors and Reviewers
Paper submitted to Materials Journal
Manuscript ID: materials-480708
Title: Preparation and characterization of high amylose corn starch/microcrystalline cellulose aerogel with high absorption
Dear Editors and Reviewers:
Thank you for your letter and for the reviewer’s comments on our manuscript entitled “Preparation and characterization of high amylose corn starch/microcrystalline cellulose aerogel with high absorption” (Materials). Those comments are all valuable and very useful to perfect our manuscript and improve our paper, as well as the important guiding significance to our researches. We have studied comments carefully and have revised the manuscript according to your suggestion which we hope to meet with approval.
We would like to express our great appreciation for your comments on our paper. We responded point by point to each reviewer comments as listed below, along with a clear indication of the location of the revision.
To Reviewer 3:
Comment 1: The authors have described the experiments, data and deductions in a clear and concise matter. What helped the authors were that the experimental data did not show any deviation or anomaly. Hence, the explanations were simple to understand and easy to deduce. Overall, I believe that this work is sound and worth publishing. Some minor editing and language correction is however required.
Authors Response: We sincerely thank you for taking time to read the article and give us your scientific comments. We have revised editing and language of the full text.
We again sincerely thank you for your time to go through this manuscript, and sincerely hope that our response is adequate. Thank you!
Best regards,
Min Wu, Ph D Associate professor
Assistant Dean of the College of Engineering
PO Box 50 China Agricultural University (East Campus) Qinghua Donglu 17, Haidian District Beijing 100083 P R of China
Tel.: 86 10 62736883, 13810328724
Fax: 86 10 62736883
Email: [email protected]
Reviewer 4 Report
In the introduction no information why authors used the microcellulose in their experiment and not nanocellulose. The nanocellulose with smaller particles or fibrils than microcellulose used in the experiment will consequence it would be not necessary to use the ball mill, is not it?
What kind of analyser machine was used to frequency sweep test?
What were the parameters of vaccum dried process? Authors wrote about temperature: -60°C and drying time as parameters, but in the methods section lack information about vaccum level and name of vaccum dried machine. The vaccum level is the important parameter during vacum drying, and probably this parameter has significiant impact for aero gels parameters.
On the 185-193 rows authors described the effect of the oil viscosity on the oil absorbing ability. At the "Materials and Methods" section, authors described the adsorption measurement methodology, only.
In mentioned part (lines: 185-193) of the manuscript, the authors used adsorption and absorption ratio. Next, authors described the results and used the absorption ratio (row 191) but on the 194 line the adsorption values are discussed. Could authors clearly write when they measured adsorprion and when and how they measured absorption. This section is unclear and needs to be rebuild.
The main quantitative results of the research should be added in Abstract, Conclusion and Highlights Sections. Authors can give concrete results illustrating the novelty of research results.
Author Response
Detailed Response to Editors and Reviewers
Paper submitted to Materials Journal
Manuscript ID: materials-480708
Title: Preparation and characterization of high amylose corn starch/microcrystalline cellulose aerogel with high absorption
Dear Editors and Reviewers:
Thank you for your letter and for the reviewer’s comments on our manuscript entitled “Preparation and characterization of high amylose corn starch/microcrystalline cellulose aerogel with high absorption” (Materials). Those comments are all valuable and very useful to perfect our manuscript and improve our paper, as well as the important guiding significance to our researches. We have studied comments carefully and have revised the manuscript according to your suggestion which we hope to meet with approval.
We would like to express our great appreciation for your comments on our paper. We responded point by point to each reviewer comments as listed below, along with a clear indication of the location of the revision.
To Reviewer 4:
Comment 1: In the introduction no information why authors used the microcellulose in their experiment and not nanocellulose. The nanocellulose with smaller particles or fibrils than microcellulose used in the experiment will consequence it would be not necessary to use the ball mill, is not it?
Authors Response: Thanks for your suggestions. In this paper, we have added the reasons of using microcrystalline cellulose in Line 57-58. In this paper, after microcrystalline cellulose ball milling for 1h, smaller particle size can be obtained to meet the experimental requirements. In the future study, we will consider the extraction of microcrystalline cellulose from agricultural waste to prepare aerogel. Compared with microcrystalline cellulose, nanocellulose is more expensive and harder to obtain
Comment 2: What kind of analyser machine was used to frequency sweep test?
Authors Response: Thanks for your suggestions. creep-recovery measurements, frequency sweep test and temperature sweep test were measured all using DMA to obtain mechanical characteristic. We have added this content to the MS. (Line 147-148 and Line 154-155 in the new revised manuscript)
Comment 3: What were the parameters of vacuum dried process? Authors wrote about temperature: -60°C and drying time as parameters, but in the methods section lack information about vacuum level and name of vacuum dried machine. The vacuum level is the important parameter during vacuum drying, and probably this parameter has significant impact for aero gels parameters.
Authors Response: Thanks for your valuable suggestion. We are very sorry that we didn’t make it clear. The aerogels were fabricated by a freeze-drying machine (LGJ-18C, China) at a temperature of−60 °C and a pressure of 15 Pa for 48 h. The detailed revision can be found in Line 92-93.
Comment 4: On the 185-193 rows authors described the effect of the oil viscosity on the oil absorbing ability. At the "Materials and Methods" section, authors described the adsorption measurement methodology, only.
Authors Response: Thanks for your suggestions. The adsorption properties of aerogel are determined by the same method regardless of oil viscosity. In this paper, the adsorption capacity of aerogel on two kinds of oil was compared to explain the influence of oil viscosity on the adsorption capacity of aerogel.
Comment 5: In mentioned part (lines: 185-193) of the manuscript, the authors used adsorption and absorption ratio. Next, authors described the results and used the absorption ratio (row 191) but on the 194 line the adsorption values are discussed. Could authors clearly write when they measured adsorprion and when and how they measured absorption. This section is unclear and needs to be rebuild.
Authors Response: Thanks for your suggestions. We are very sorry for our negligence. The words “absorption rate” have been changed into “absorption ratio”. The method for measuring adsorption is given in lines 125-131.
Comment 6: The main quantitative results of the research should be added in Abstract, Conclusion and Highlights Sections. Authors can give concrete results illustrating the novelty of research results.
Authors Response: Thanks for your suggestions. We have revised the abstract and conclusion in the revised manuscript.
We again sincerely thank you for your time to go through this manuscript, and sincerely hope that our response is adequate. Thank you!
Best regards,
Min Wu, Ph D Associate professor
Assistant Dean of the College of Engineering
PO Box 50 China Agricultural University (East Campus) Qinghua Donglu 17, Haidian District Beijing 100083 P R of China
Tel.: 86 10 62736883, 13810328724
Fax: 86 10 62736883
Email: [email protected]
Round 2
Reviewer 2 Report
The manuscript “Preparation and characterization of high amylose corn starch/microcrystalline cellulose aerogel with high absorption” has been improved over the previous version, but some minor changes have yet to be made:
- P. 3, L. 108. Please remove “purification” from “purification ?0”! The explanation must be changed with “?0 – is bulk density and ? - is the skeletal density”! What value did the authors used for ??
- P. 3, L. 118. Please change the explanation “I002 denotes the amorphous and crystalline regions, while Iamorph relates to the amorphous fraction” with the following: “I002 is the maximum intensity of the (002) lattice diffraction and Iamorph is the intensity diffraction of the amorphous band”! This is an empirical method proposed by Segal et al.!
- P. 4, L.128. The sentence is still not clear “the aerogel was salvaged to make it free to drop 30s and weigh the quality at this time”! Please change it with “the aerogel was wiped with a soft tissue for 30s to remove surface oil, and then was weighed”, if the explanation is correct!
- P.5, L 172-175. The sentence is repetitive: “However, as more amount of starch was added, the density reduced greatly and porosity increased significantly (P < 0.05) compared with S0CA, perhaps because with the increase of starch content resulted in the decrease of the content of microcrystalline cellulose, so the density decreased, and porosity increased”! Please revise the explanation!
- P.5, L 176. What did the authors want to say: “Vacuum freeze drying is heating in a vacuum environment”? In my opinion this sentence should be removed!
- P.5, L. 177. I think that the sentence “to reduce damage” should be removed! There is no damage in the preparation of the aerogel!
- P.5, L. 181. “starch as crosslinking agent”?? The starch is not a cross-linking agent!! It was done a blend between cellulose and starch and no cross-linking agent was used! Please revise the terminology throughout the text!
- P.6, L. 213. There is no need to repeat the sentence “with the increase of starch content and the decrease of microcrystalline cellulose content”, because it is understandable that by increasing the starch content, the cellulose content decreases! Please make the correction within manuscript in order to avoid burdening explanation!
- P. 7, L. 223-229. The sentences should be changed with: “The XRD patterns of obtained aerogels with different starch addition, as well as of starch and of microcrystalline cellulose balled-milled for 1h are shown in Fig. 3. The X-ray diffractogram of microcrystalline cellulose ball-milled for 1h is characteristic to cellulose I crystal structure (Fig. 3), according to literature data [28]. In case of aerogels, the corresponding X-ray diffraction patterns are characteristic to cellulose II, due to the dissolving/regeneration process of microcrystalline cellulose in NaOH/urea aqueous system. As we can see for all aerogels, the characteristic diffraction peaks are observed at an Bragg angle of 12°, 20° and 22°, and are assigned to (110), (110) and (200) crystallographic planes, respectively.”!
- P.7, L. 234-235. The sentence “The crystallinity decreases because the surface area corresponding to the amorphous region increases” should be “The crystallinity of the sample decreases due to the addition within aerogels of starch, with a high content of amorphous region.”!
Author Response
Detailed Response to Editors and Reviewers
Paper submitted to Materials Journal
Manuscript ID: materials-480708
Title: Preparation and characterization of high amylose corn starch/microcrystalline cellulose aerogel with high absorption
Dear Editors and Reviewers:
Thank you very much for your patience and valuable help in our manuscript entitled “Preparation and characterization of high amylose corn starch/microcrystalline cellulose aerogel with high absorption” (Materials). Your comments and suggestions are all valuable and very useful to perfect our manuscript and improve the quality. We have revised the manuscript according to your suggestion and submitted it. Attached please find the revised version, which we hope to meet with approval.
We would like to express our great appreciation for your comments on our paper. The main corrections in the paper and the responses to your comments are as follows, along with a clear indication (green) of the location of the revision.
To Reviewer 2:
The manuscript “Preparation and characterization of high amylose corn starch/microcrystalline cellulose aerogel with high absorption” has been improved over the previous version, but some minor changes have yet to be made:
Comment 1: P. 3, L. 108. Please remove “purification” from “purification ?0”! The explanation must be changed with “?0 – is bulk density and ? - is the skeletal density”! What value did the authors used for ??
Authors Response: Thanks for your suggestions, we appreciate it very much. We have now revised and refined the explanation of ? and ?0 according to your suggestions in Line 106 and Line 108. The skeletal density ? have been calculated by Eq. (2) with different starch content.
Comment 2: P. 3, L. 118. Please change the explanation “I002 denotes the amorphous and crystalline regions, while Iamorph relates to the amorphous fraction” with the following: “I002 is the maximum intensity of the (002) lattice diffraction and Iamorph is the intensity diffraction of the amorphous band”! This is an empirical method proposed by Segal et al.!
Authors Response: Thanks for your suggestions, we have now revised and refined explanation of I002 and Iamorph according to your suggestions in Line 119-120. In addition, we have added a literature written by Segal et al.
Comment 3: P. 4, L.128. The sentence is still not clear “the aerogel was salvaged to make it free to drop 30s and weigh the quality at this time”! Please change it with “the aerogel was wiped with a soft tissue for 30s to remove surface oil, and then was weighed”, if the explanation is correct!
Authors Response: We sincerely thank you give us your advice. We have changed the sentence with “then the aerogel was put on the sieve for 30s to remove surface oil, and then was weighed” in Line 127-128.
Comment 4: P.5, L 172-175. The sentence is repetitive: “However, as more amount of starch was added, the density reduced greatly and porosity increased significantly (P < 0.05) compared with S0CA, perhaps because with the increase of starch content resulted in the decrease of the content of microcrystalline cellulose, so the density decreased, and porosity increased”! Please revise the explanation!
Authors Response: We sincerely thank you give us your advice. We have changed the explanation with “because of the decrease of the content of microcrystalline cellulose” in Line 173-174.
Comment 5: P.5, L 176. What did the authors want to say: “Vacuum freeze drying is heating in a vacuum environment”? In my opinion this sentence should be removed!
Authors Response: We sincerely thank you give us your advice. Now we have removed the sentence according to your suggestion and we have changed our expression in Line 177-178.
Comment 6: P.5, L. 177. I think that the sentence “to reduce damage” should be removed! There is no damage in the preparation of the aerogel!
Authors Response: Thanks for your suggestions, we have now removed the sentence “to reduce damage”.
Comment 7: P.5, L. 181. “starch as crosslinking agent”?? The starch is not a cross-linking agent!! It was done a blend between cellulose and starch and no cross-linking agent was used! Please revise the terminology throughout the text!
Authors Response: We are very sorry for our negligence. We have emended it in our revised manuscript. We have removed the content about the crosslinking between starch and MCC.
Comment 8: P.6, L. 213. There is no need to repeat the sentence “with the increase of starch content and the decrease of microcrystalline cellulose content”, because it is understandable that by increasing the starch content, the cellulose content decreases! Please make the correction within manuscript in order to avoid burdening explanation!
Authors Response: Thanks for your suggestions, we have now removed the sentence “and the decrease of microcrystalline cellulose content”. And we have changed the expression throughout the text.
Comment 9: P. 7, L. 223-229. The sentences should be changed with: “The XRD patterns of obtained aerogels with different starch addition, as well as of starch and of microcrystalline cellulose balled-milled for 1h are shown in Fig. 3. The X-ray diffractogram of microcrystalline cellulose ball-milled for 1h is characteristic to cellulose I crystal structure (Fig. 3), according to literature data [28]. In case of aerogels, the corresponding X-ray diffraction patterns are characteristic to cellulose II, due to the dissolving/regeneration process of microcrystalline cellulose in NaOH/urea aqueous system. As we can see for all aerogels, the characteristic diffraction peaks are observed at an Bragg angle of 12°, 20° and 22°, and are assigned to (110), (110) and (200) crystallographic planes, respectively.”!
Authors Response: Thanks for your suggestions, we have now revised and refined our sentences and paragraphs according to your suggestions. Now we have changed this content to the MS. (Line 220-227 in the new revised manuscript).
Comment 10: P.7, L. 234-235. The sentence “The crystallinity decreases because the surface area corresponding to the amorphous region increases” should be “The crystallinity of the sample decreases due to the addition within aerogels of starch, with a high content of amorphous region.”!
Authors Response: We sincerely thank you give us your advice. We have now revised and refined our sentences and paragraphs according to your suggestions in Line 233-234.
We again sincerely thanks for your patience and careful guidance, and sincerely hope that our response is adequate. Thank you!
Best regards,
Min Wu, Ph D Associate professor
Assistant Dean of the College of Engineering
PO Box 50 China Agricultural University (East Campus) Qinghua Donglu 17, Haidian District Beijing 100083 P R of China
Tel.: 86 10 62736883, 13810328724
Fax: 86 10 62736883
Email: [email protected]
Reviewer 4 Report
Dear Authors,
thank you for improving this manuscript. I approve and accept all correction made in the text by Authors. Ihe paper is suitable for publication.
Author Response
Detailed Response to Editors and Reviewers
Paper submitted to Materials Journal
Manuscript ID: materials-480708
Title: Preparation and characterization of high amylose corn starch/microcrystalline cellulose aerogel with high absorption
Dear Editors and Reviewers:
Thank you very much for your patience and valuable help in our manuscript entitled “Preparation and characterization of high amylose corn starch/microcrystalline cellulose aerogel with high absorption” (Materials). Your comments and suggestions are all valuable and very useful to perfect our manuscript and improve the quality. We have revised the manuscript according to your suggestion and submitted it. Attached please find the revised version, which we hope to meet with approval.
We would like to express our great appreciation for your comments on our paper. The main corrections in the paper and the responses to your comments are as follows, along with a clear indication of the location of the revision.
To Reviewer 4:
Comment 1: Thank you for improving this manuscript. I approve and accept all correction made in the text by Authors. The paper is suitable for publication.
Authors Response: We sincerely thank you for taking time to read the article and give us your scientific comments.
We again sincerely thanks for your patience and careful guidance, and sincerely hope that our response is adequate. Thank you!
Best regards,
Min Wu, Ph D Associate professor
Assistant Dean of the College of Engineering
PO Box 50 China Agricultural University (East Campus) Qinghua Donglu 17, Haidian District Beijing 100083 P R of China
Tel.: 86 10 62736883, 13810328724
Fax: 86 10 62736883
Email: [email protected]